

# *Melanella martarum* sp. nov. (Gastropoda: Eulimidae): the first parasitic deep-sea snail reported for the Salas & Gomez Ridge

Leonardo Santos de Souza[1,*], Cynthia M. Asorey[2,3,*] and Javier Sellanes[2,3]

[1] Departamento de Zoologia - Instituto de Biociências, Universidade Federal do Rio Grande do Sul (UFRGS), Rio Grande do Sul, Brazil

[2] Departamento de Biologia Marina, Center for Ecology and Sustainable Management of Oceanic Islands (ESMOI), Coquimbo, Chile

[3] Sala de Colecciones Biológicas Universidad Católica del Norte (SCBUCN), Coquimbo, Chile

[*] These authors contributed equally to this work.

Corresponding author
Cynthia M. Asorey,
cynthiaasorey@gmail.com

## ABSTRACT

Eulimidae is a highly diverse family of gastropods that are often parasites of echinoderms. They are cosmopolitan and live from the intertidal to great depths. Despite its wide geographic and bathymetric distribution, no species of Eulimidae have been reported for the Salas & Gómez Ridge to date. In this study, we describe *Melanella martarum* sp. nov., which was collected during the EPIC oceanographic cruise onboard RV Mirai (JAMSTEC, Japan) in 2019. Seven specimens were collected with a modified Agassiz trawl on the summit of seamount "Pearl" (Zhemchuznaya) in the Salas & Gómez Ridge (25.59°S, 89.13°W) at 545 m depth. The morphology of *M. martarum* sp. nov. was compared with other *Melanella* species reported for the area, including Chile and Rapa Nui. DNA was extracted and partial sequences of the mitochondrial genes Cytochrome Oxidase 1 (COI) and 16S rDNA, and the nuclear gene Histone 3 (H3) were sequenced. *Melanella martarum* sp. nov. has morphological characteristics that separate it from other species of *Melanella*, such as the thickness and color of the shell, and the shape of the protoconch. In addition, *M. martarum* sp. nov. was genetically differentiated from other *Melanella* spp. sequences (uncorrected p distances from 18,1–8.6% in mitochondrial COI and 16S rDNA to 3% in nuclear H3 sequences). Although there is not much molecular data available for Eulimidae, the phylogenetic analysis confirms the results obtained by morphology, placing the species found on the Salas & Gómez Ridge within the genus *Melanella*. The current study advances the understanding of the poorly known benthic fauna found on seamounts in the easternmost part of the Sala & Gómez ridge, a location distinguished by a high level of endemism.

## INTRODUCTION

Eulimidae Philippi, 1853 is a family of marine gastropods that encompass parasites of all extant classes of Echinodermata, presenting a high species richness with diverse

body plans and parasitic strategies (*Warén, 1983*; *Takano & Kano, 2014*; *Takano & Goto, 2021*). Currently, there are around 960 valid species (*MolluscaBase, 2023*), but *Warén & Gittenberger (1993)* provided a gross estimate that there could be more than 4,000 species of Eulimidae. Other authors also highlighted the underestimated diversity of this group. For example, *Bouchet et al. (2002)* based on a huge sampling effort in New Caledonia estimated that 80% of the eulimids collected may represent undescribed species.

The lack of knowledge about Eulimidae is not only related to the inventory of species but also to the parasitic strategies of this group of snails. Some eulimids are strongly attached to their hosts (*i.e.,* tightly attached ectosymbionts, gall-forming species, and endoparasites) but also free-living ectosymbionts (*Dgebuadze et al., 2022*). In the former case, these snails can detach from the host more easily when disturbed during sampling with common benthic tools (nets, grabs, dredges), this often makes it difficult to identify the hosts (*Takano, Itoh & Kano, 2018*; *Takano, Kimura & Kano, 2020*).

The genus *Melanella* is one of the most species-rich of the family, with more than 200 recognized species (*MolluscaBase, 2023*). *Melanella* is known to live as an ecto- or endoparasite of holothuroids in the orders Holothuriida (formerly Aspidochirotida) and Dendrochirotida (*Warén, 1983*) and species can be host-specific or generalist (*Crossland, Alford & Collins, 1993*).

In Chile, 15 species of eulimid gastropods have been reported (10 genera) (*Rehder, 1980*; *Valdovinos, 1999*; *Osorio, 2023*). Most of the species described so far are found on Rapa Nui Island (Easter Is.) (*Linse, 1999*; *Valdovinos, 1999*; *Osorio, 2023*). *Melanella* is the most representative genus with four species: *M. cumingii* (A. Adams 1854), *M. aciculata* (Pease, 1861) and *M. pisinna* (*Rehder, 1980*) (*Rehder, 1980*; *Osorio, 2023*) found in Rapa Nui Island and *M. subantartica* (Strebel 1908) endemic of the Magellan region (*Linse, 1999*; *Valdovinos, 1999*).

Between 1973 and 1987, research expeditions from the former Soviet Union explored 22 seamounts of the Salas & Gómez Ridge (SGR) and the Nazca Ridge (NR) outside the Chilean Exclusive Economic Zone (CEEZ) (mainly west of ~83°W) (*Parin, Mironov & Nesis, 1997*). These expeditions represent only ~3% of the seamounts that make up both submarine ridges. CIMAR 22 was the first multidisciplinary expedition to study the summit of several seamounts of Salas & Gómez Island and Desventuradas Islands within the CEEZ (*Tapia-Guerra et al., 2021*). This new expedition has added new records and/or new species for science, mainly including crustaceans, echinoderms, polychaeta, and mollusks (*Mecho et al., 2019*; *Sellanes et al., 2019*; *Asorey et al., 2020*; *Díaz-Díaz et al., 2020*; *Gallardo et al., 2021*). However, no specimens of the familyEulimidae have been collected in this expedition. Furthermore, in the studies carried out outside the CEEZ, no new species have been described or the presence of any specimen of this gastropod family has been reported. But in 2019, during the Japan Agency for Marine-Earth Science and Technology (JAMSTEC) oceanographic cruise, some specimens of this family were collected from the seamount ''Pearl'' (Zhemchuznaya). So, in the present study, we describe these samples as the first deep-sea parasitic snail species of Eulimidae for the SGR. We also provide genetic data of the new species, assessing its phylogenetic relationships with congeners, as well as insight into its echinoderm host.

**Table 1** Voucher and shell measurements of type specimens of *Mellanella martarum* sp. nov.

| Specimen | Voucher | Shell high (mm) | Shell width (mm) | Aperture height (mm) | Aperture width (mm) | Number of Teleconcha Whorls |
|---|---|---|---|---|---|---|
| *Mellanella martarum sp. nov.* | SCBUCN-8611-1 | 4.61 | 1.85 | 1.35 | 0.75 | 5 |
| *Mellanella martarum sp. nov.* | SCBUCN-8611-2 | 4.42 | 1.75 | 1.32 | 0.88 | 5 |
| *Mellanella martarum sp. nov.* | SCBUCN-8611-3 | 4.50 | 2.00 | 1.4 | 0.77 | 5 |
| *Mellanella martarum sp. nov.* | SCBUCN-8612 | 4.65 | 2.10 | 1.26 | 0.85 | 5 |
| *Mellanella martarum sp. nov.* | MNHNCL-205422 | 3.76 | 1.65 | 1.36 | 0.68 | 4 |
| *Mellanella martarum sp. nov.* (Holotype) | MNHNCL-205421 | 6.25 | 2.45 | 1.7 | 1.03 | 6 |
| [*] *Mellanella martarum sp. nov.* | SCBUCN-5482 | 8.14 | 2.39 | NA | NA | 9 |

Notes.
[*]Sample used by SEM images and molecular analysis.

# MATERIALS & METHODS

## Material collection and sampling site

Samples were obtained by a modified Agassiz trawl with a mouth of 1.5 m × 0.5 m (width × height) fitted with a net of 12 mm mesh at the cod end and operated in 10 min hauls (bottom contact) at ~3 knots during the oceanographic EPIC cruise onboard the RV Mirai (JAMSTEC, Japan) in 2019. Sampling was performed on the summit of the seamount "Pearl" (Zhemchuznaya, 25.59°S, 89.13°W), at 545 m depth, in SGR. The collected material was preserved in 95% ethanol. Holotype and paratype specimens are housed in Museo Nacional de Historia Natural (MNHNCL) and Sala de Colecciones Biológicas de la Universidad Católica del Norte (SCBUCN), both in Chile (Table 1). Sample collection was performed under the permission of Res. Ext No. 3685/2016 from SUBPESCA (Chile) to Universidad Católica del Norte.

## Phylogenetic analysis

Genomic DNA was extracted from the whole animal of SCBUCN-5482 using an E.Z.N.A.® Tissue DNA kit (Omega, Bio-Tek, Norcross, GA, USA). To amplify partial sequences of the Histone 3 (H3) nuclear gene and the mitochondrial cytochrome C oxidase I (COI) and 16S rRNA genes, the pairs of primers H3F and H3R (*Colgan, Ponder & Eggler, 2000*), HCO-1490 and LCO-2198 (*Folmer et al., 1994*), and 16SAR-16SBR (*Palumbi et al., 1991*) were used, respectively. The PCR profile for COI started with 5 min at 95 °C, followed by 40 cycles of denaturation at 95 °C (1 min), annealing at 50 °C (1 min), and elongation at 72 °C (2 min), with a final elongation phase at 72 °C (13 min). A similar PCR profile was set for H3 and 16S rRNA (annealing at 55 °C). The resulting amplicons were visualized in agarose 1% gels and the PCR products were sent to Macrogen, Inc. (Seoul, South Korea) for DNA Sanger sequencing.

The COI, 16S rRNA, and H3 sequences (forward and reverse) were visualized and assembled with Geneious Prime 2022. 2.2 (*Kearse et al., 2012*). Fifteen H3 and 16S rRNA genes and fourteen COI gene sequences of Eulimidae were extracted from NCBI GenBank (Table 2), concatenated, and aligned with the ones of *Melanella martarum* sp. nov., using

**Table 2** **Species used in present analyses with GenBank accession numbers and the authors of each one.** Accession numbers of newly obtained sequences are given in bold.

| Family | Species | H3 | 16S | COI | Published by |
|---|---|---|---|---|---|
| Eulimidae | *Hemiaclis sp.* | AB930436 | AB930409 | AB930465 | *Takano & Kano (2014)* |
| | *Hemiliostraca sp.* | AB930437 | AB930410 | AB930466 | *Takano & Kano (2014)* |
| | *Melanella acicula* | AB930435 | AB930408 | AB930464 | *Takano & Kano (2014)* |
| | *Monogamus entopodia* | AB930429 | AB930402 | AB930458 | *Takano & Kano (2014)* |
| | *Niso matsumotoi* | AB930440 | AB930413 | AB930469 | *Takano & Kano (2014)* |
| | *Pyramidelloides angustus* | AB930441 | AB930414 | AB930470 | *Takano & Kano (2014)* |
| | *Stilifer akahitode* | AB930432 | AB930405 | AB930461 | *Takano & Kano (2014)* |
| | *Thyca crystallina* | AB930431 | AB930404 | AB930460 | *Takano & Kano (2014)* |
| | *Vitreolina aurata* | AB930428 | AB930401 | AB930457 | *Takano & Kano (2014)* |
| | *Asterophila perknasteri 1* | MN224387 | MN224427 | MN224306 | *Layton, Rouse & Wilson (2019)* |
| | *Asterophila perknasteri 2* | MN224369 | MN224437 | MN224310 | *Layton, Rouse & Wilson (2019)* |
| | *Asterophila sp 4 KKSL-2019* | MN224372 | MN224460 | MN224348 | *Layton, Rouse & Wilson (2019)* |
| | *Asterophila perknasteri 3* | MN224388 | MN224451 | MN224362 | *Layton, Rouse & Wilson (2019)* |
| | *Melanella Sp. CKC-2011* | JF750989 | JF750955 | – | *Churchill, Strong & Foighil (2011)* |
| | *Fusceulimoides kohtsukai* | LC726229.1 | LC726230.1 | LC726231.1 | *Takano et al. (2023)* |
| | ***Melanella martarum* sp. nov.** | **OP589975** | **OP575953** | **OP577852** | **This study** |
| Vanikoridae | *Vanikoro helicoidea* | AB930450 | AB930421 | AB930487 | *Takano & Kano (2014)* |

default MUSCLE (*Edgar, 2004*) parameters. The resulting alignment was used to construct the maximum likelihood phylogenetic tree with the RAxML 8.2.11 software (*Guindon et al., 2010*) plugin for Geneious Prime 2022.2.2 (*Kearse et al., 2012*), using the following settings: Nucleotide model = GTR GAMMA, Algorithm = Rapid bootstrapping and search for best-scoring ML tree, bootstrap replicates = 1,000, Partitioning = 16S = 1-419, COI = 420–1,049, H3 = 1,050–1,363. Significant bootstrap values (>90) are reported at the nodes. Sequences of the 3 above-mentioned markers of *Vanikoro helicoidea* (Vanikoridae) were used as an outgroup.

To check whether the phylogenetic relationships of *Melanella martarum* sp. nov are consistent with the increase in species, 36 COI gene sequences of Eulimidae were extracted from NCBI GenBank (Table S1, Supplementary Material) and were aligned with *Melanella martarum* sp. nov., using default MUSCLE (*Edgar, 2004*) parameters. The resulting alignment was used to construct the maximum likelihood phylogenetic tree with the RAxML 8.2.11 software (*Guindon et al., 2010*) plugin for Geneious Prime 2022.2.2 (*Kearse et al., 2012*), using the following settings: Nucleotide model = GTR GAMMA I, Algorithm = Rapid bootstrapping and search for best-scoring ML tree, bootstrap replicates = 1,000, Partitioning = DNA, gene1codon1 = 1-630\3,

DNA, gene1codon2 = 2-630\3, DNA, gene1codon3 = 3-630\3. The COI sequence of *Vanikoro helicoidea* (Vanikoridae) AB930487 was used as an outgroup.

## SEM images

The shell morphology was examined with a Hitachi SU3500 scanning electron microscope (SEM) at the Microscopy Laboratory of the Facultad de Ciencias del Mar, Universidad

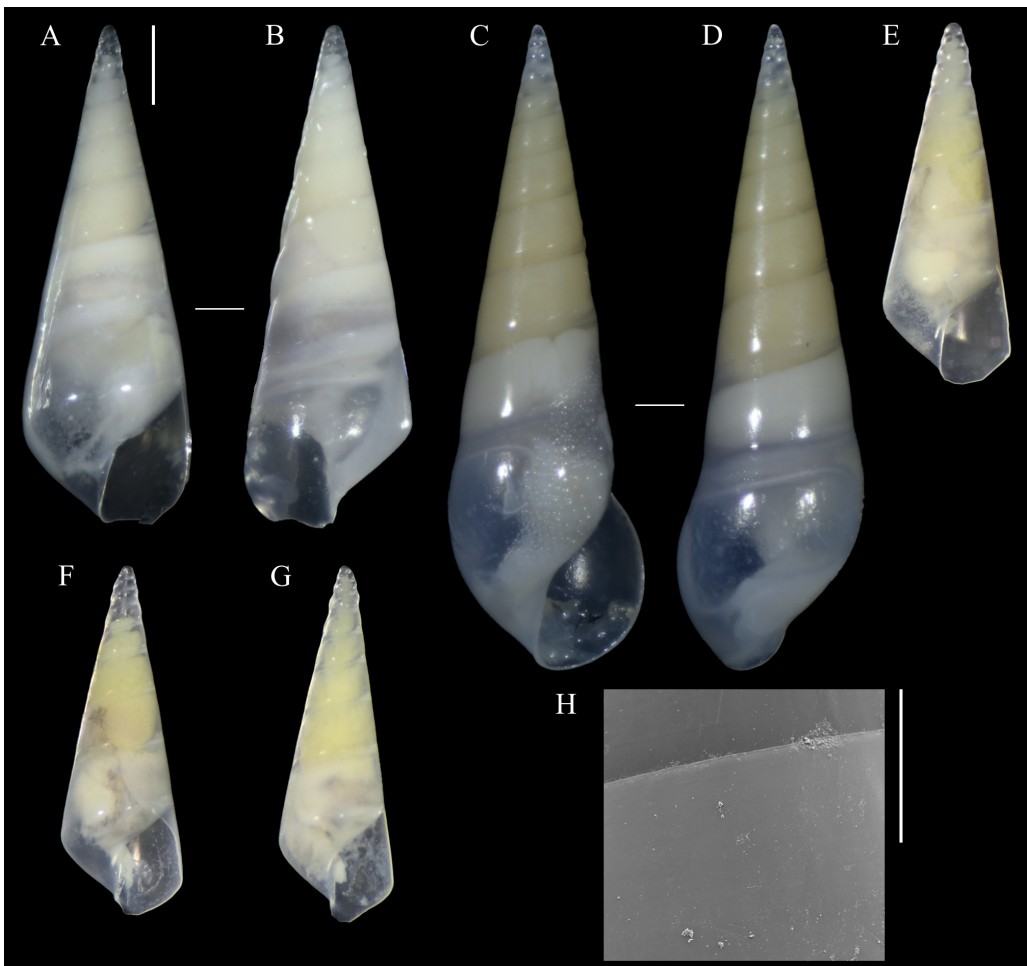

**Figure 1** *Melanella martarum.* **sp. nov.** (A, B) Holotype, MNHNCL-205421. (C, D, H). Paratype, SCBUCN 5482. (E–G). Paratypes, SCBUCN 8611. (A, C, E–G). Frontal view. (B, D). Dorsal view. (H). Detail of teleoconch surface, close to suture. Scale bars: (A–G) = one mm (entire shells at same scale); $H = 100\,\mu m$.

Católica del Norte, Coquimbo, Chile. The shell was dried in a Tousimis, Samdri-780A critical-point dryer using CO2, mounted on bronze stubs, and coated with gold in a JEOL JFC-100 evaporator. The examined individual was from the same specimen used for the molecular analysis (SCBUCN 5482).

## Shell measurements

Shell measurements follow *Souza & Pimenta (2019*: fig. 1): SL: shell length; SW: shell width; BWL: ultimate whorl length; AL: aperture length; AW: aperture width; PCH: protoconch height. The spire angle was measured through images of the shell in frontal view, with the vertex centralized at the apex and pointing the arcs to the sutures at both sides of the ultimate whorl.

**New species registration**

The following information was supplied regarding the registration of a newly described species:

Publication LSID: urn:lsid:zoobank.org:pub:716D01AC-7DA7-4B2D-A4D5-0DD925825BC6

*Melanella martarum* sp. nov. LSID: urn:lsid:zoobank.org:act:CC4C4BE5-4D7F-4EF3-8926-9B960C01C3CA

## RESULTS

### Systematic account

Family EULIMIDAE Philippi, 1853
Genus *Melanella* Bowdich, 1822

Type species: *Melanella dufresnii* Bowdich, 1822, by monotypy.

*Melanella martarum* sp. nov.
Figs. 1A–1H, 2A–2C–2K

**Diagnosis:** Shell medium in size (up to 8.1 mm long), polished, translucent white, conical, apex minute (Fig. 1). Protoconch multispiral of about 4 whorls (Fig. 2), with slightly convex whorls, surface smooth, with no distinction between protoconch I and II; transition to teleoconch marked by sinuous incremental scar. Teleoconch whorls almost flat, incremental scars irregularly spaced, surface smooth. Ultimate whorl about 45% of total shell length in adults, base rounded. Aperture wide, squarish in juveniles, pear-shaped and laterally expanded in adults; outer lip sinuous. Eyes present, anterior body whitish, posterior region orange. Ectoparasitic on holothuroids.

**Description (Holotype):** Shell conical, apex obtuse, reaching about 6.3 mm long, 2.5 mm wide, spire angle 26°. Larval shell vitreous, about 4.0 convex whorls, 500 μm in height, smooth, transition to teleoconch marked by distinct incremental scar. Teleoconch vitreous, not colored, about 6.5 whorls of flat outline; suture shallow, slightly impressed, sloping; subsutural zone about 1/5 of the height of the whorl; surface glossy, smooth, except for incremental scars appearing at irregular intervals. Last whorl about 50% of the shell length; base slightly rounded. Aperture high, 60% of ultimate whorl height, rhomboid in shape, acute above, slightly rounded below; outer lip thin, sinuous, opisthocline, retracted near the suture, maximum projection at the middle of the outer lip height; inner lip thin, sinuous, sloping. Not umbilicate.

**Description (Paratype SCBUCN-5482):** Shell conical, apex obtuse, reaching about 8.1 mm long, 2.6 mm wide, spire angle 20°. Larval shell vitreous, about 4.0 convex whorls, 480 μm in height, smooth, transition to teleoconch marked by distinct incremental scar.

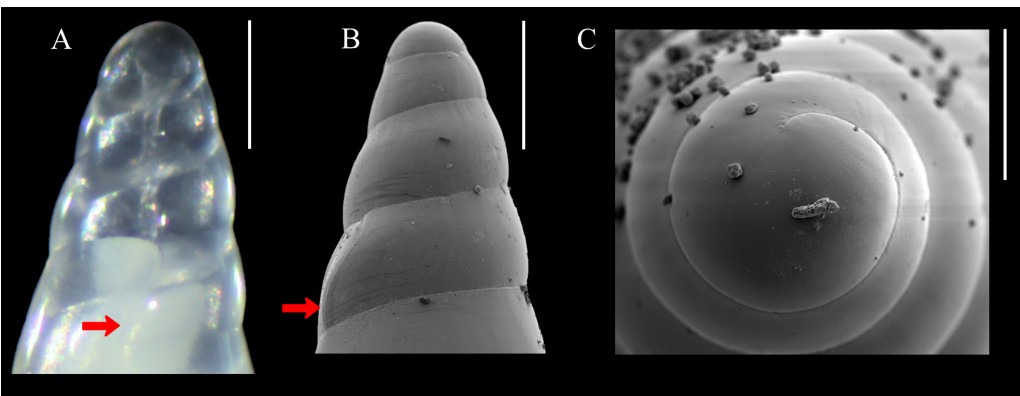

**Figure 2** *Melanella martarum.* **sp. nov.** (A) SCBUCN 8542, detail of protoconch, frontal view. (B–C). Paratype, SCBUCN 5482, detail of protoconch, frontal and apical views, respectively. Arrows in (A) and (B) indicate transition protoconch-teleoconch. Scale bars: (A–B) = 200 μm; C = 100 μm.

Teleoconch vitreous, not colored, about 9.0 whorls of flat outline; suture shallow, slightly impressed, sloping; subsutural zone about 1/5 of the height of the whorl; surface glossy, smooth, except for incremental scars appearing at irregular intervals. Last whorl about 45% of the shell length; base rounded. Aperture high, 65% of ultimate whorl height, pear-shaped, moderately expanded laterally, acute posteriorly, rounded and spread anteriorly; outer lip thickened, very sinuous, opisthocline, retracted near the suture, after strongly projecting, and retracted in the distal region, maximum projection at the middle of the outer lip height; inner lip thin, sinuous, sloping. Not umbilicate.

**Holotype: MNHNCL-205421**, SL: 6.3 mm, SW: 2.5 mm, AL: 1.7 mm, AW: 1.0 mm, Teleoconch whorls: 6.5. Seamount ''Pearl'' off the coast of Chile in international waters, 25.59°S, 89.13°W, 545 m depth, February 9th, 2019, RV *Mirai* (JAMSTEC, Japan).

**Paratypes (all from type locality):** SCBUCN 8611-1, SL: 4.6 mm, SW: 1.9 mm, AL: 1.4 mm, AW: 0.8 mm, Teleoconch whorls: 5; SCBUCN 8611-2, SL: 4.4 mm, SW: 1.8 mm, AL: 1.3 mm, AW: 0.9 mm, Teleoconch whorls: 5; SCBUCN 8611-3, SL: 4.5 mm, SW: 2.0 mm, AL: 1.4 mm, AW: 0.8 mm, Teleoconch whorls: 5; SCBUCN 8612-1, SL: 4.7 mm, SW: 2.1 mm, AL: 1.3 mm, AW: 0.9 mm, Teleoconch whorls: 5; MNHNCL-205422, SL: 3.8 mm, SW: 1.7 mm, AL: 1.4 mm, AW: 0.7 mm, Teleoconch whorls: 4; SCBUCN-5482, SL: 8.1 mm, SW: 2.9 mm, AL: 2.2 mm, AW: 1.4 mm, Teleoconch whorls: 9.

**Comparative material (examined through photographs):** *Melanella aciculata* (Pease, 1861) (Fig. 3): Lectotype NHMUK 1962839 (Fig. 3A) (designated by *Kay, 1965*), Sandwich Islands, Hawaiian Archipelago; Paralectotypes NHMUK 1962840 (Figs. 3C–3F), 4 shells, from type locality; Paralectotype MCZ 31705 (Figs. 3G–3H), 1 shell, Sandwich Islands; Paralectotype MCZ 187747, 1 shell, Hawaiian Islands. *Melanella acicula* (A. Gould, 1849): Syntype ANSP 19773 of *Eulima pisorum* (*Pilsbry, 1917*) [= *M. acicula*], 1 shell, Viti Islands, Fiji (see http://clade.ansp.org/malacology/collections/search.php?submitbut= Search&name=Eulima+pisorum&location=&agent=&catalog=). *Melanella micans* (P.P. Carpenter, 1865): Holotype USNM 14850 of *Eulima micans* P.P. Carpenter, 1865, San

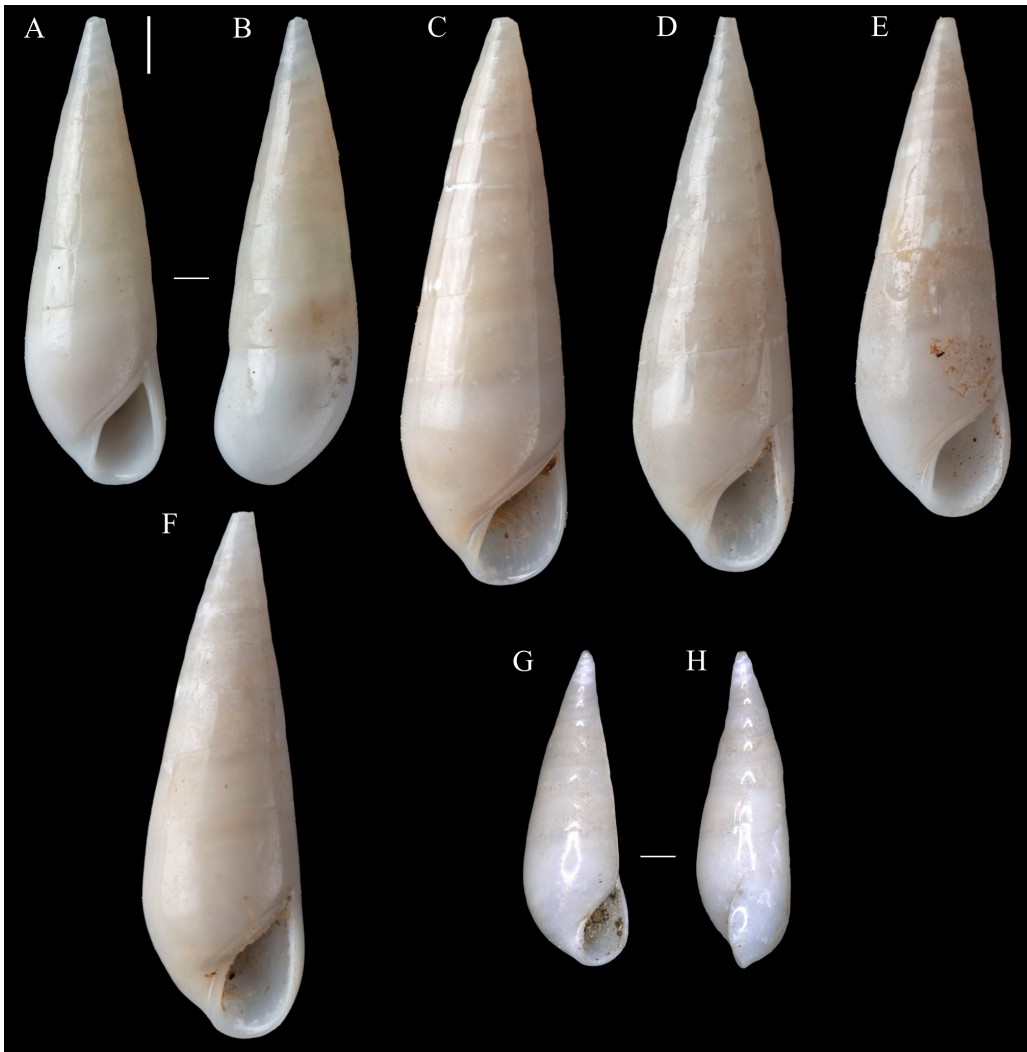

**Figure 3** *Melanella* **aciculata (Pease, 1861).** (A, B) Lectotype, NHMUK 1962839. (C–F) Paralectotypes, NHMUK 1962840A–D. (G, H) Paralectotype, MCZ 3170. (A, C–G). Frontal view. (B) Dorsal view. (H) Lateral view. Scale bar: (A–H) = one mm (shells at same scale). Credits: Images (A–F), courtesy of Andreia Salvador and photographic unit (NHMUK); (G, H): courtesy of Jennifer W. Trimble (MCZ).

Pedro, California, USA; Holotype USNM 267304 of *Melanella mexicana* (*Bartsch, 1917*) (junior synonym)], Gulf Coast of Loewer California. *Polygireulima rutila* (P.P. Carpenter, 1864): Holotype USNM 14828 of *Eulima rutila* P.P. Carpenter, 1864, Monterey, California, USA (see http://n2t.net/ark:/65665/38a09710d-1739-4a50-af42-19db3ccb960b).

**Distribution, habitat, and parasitic strategy:** *Melanella martarum* sp. nov. was found tightly attached to *Oneirophanta* cf. *setigera* (Ludwig, 1893) (Holothuroidea: Deimathidae) in general in the area near the base of the introvert (Fig. 4A). The Holothuroid host was found on a hard substrate covered with a finer, sandy-type sediment (Fig. 3B) at a depth of 545 m on the summit of a seamount in international waters off Chile (25.59°S, 89.13°W). This seamount is known as "Pearl". Ectosymbiont.

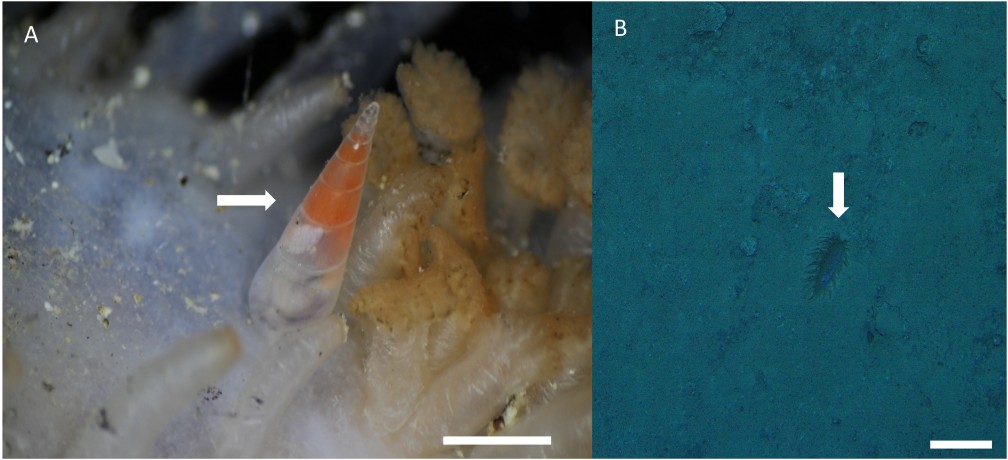

**Figure 4** *Melanella martarum.* **sp. nov. in its holothurian host.** (A) *Melanella martarum* sp. nov. SCBUCN-5482 on its holothurian host. Scale bar: four mm. The white arrow shows *M. martarum* sp. nov attached to its holothurian host. (B) Screenshot of the camera DeepTow from SPG5 seamount. Scale bar: 10 cm. The white arrow shows the host on a sandy substrate.

**Etymology:** The species name honors Marta B. Parodi and Marta S. Wainstein, grandmother and mother, respectively, of one of the authors (C.M. Asorey). The Latin genitive case suffix -arum was added as the epithet represents two females.

**Species comparison:** A few species of Eulimidae are known from this area, including *Melanella*, and most of them are known from shallow waters. *Melanella martarum* sp. nov. resembles the shallow water species *M. aciculata* (Figs. 3A–3H), an ectoparasitic species known from the Hawaiian Archipelago, French Polynesia, and Rapa Nui (*Osorio, 2018*; *Osorio, 2023*). However, the newly described species differs from *M. aciculata* by the straighter spire, a consequence of the irregular position of the incremental scars (*Bouchet & Warén, 1986*: 312) in *M. martarum* sp. nov. The initial whorls of *M. aciculata* have a more regular position of the incremental scars, giving a distorted appearance (Figs. 3G, 3H). Besides that, *M. martarum* sp. nov has an aperture more expanded laterally, a thinner inner lip and outer lip (Fig. 1C), and the adult of *M. martarum* sp. nov. has a sinuous outer lip (Figs. 1C–1D), while the largest shells of *M. aciculata* have a straighter outer lip (Figs. 3A, 3C–3F). The young holotype of *M. martarum* sp. nov. presents a spire angle of 26°, similar to the remaining young specimens (26–27°) and the adult types of *M. aciculata* (25°; $n = 5$, Figs. 3A–3F). However, the adult specimen of *M. martarum* sp. nov. shows a more acute spire angle (20°), which is possibly related to a differentiation of increment of diameter during growth.

  *Melanella martarum* sp. nov. also resembles *M. acicula*, with a broad distribution in shallow waters of the West Pacific, but the tip of the shell is much more acute in the newly described species. *Pilsbry (1917)* highlighted the broad apex of *Eulima pisorum Pilsbry, 1917*, a junior synonym of *M. acicula*, while comparing this species with *M. aciculata*. Our comparison with a syntype of *E. pisorum* also reinforces this diagnostic feature. The spire angle of the syntype ANSP 19773 of *E. pisorum*, the one with images available (remaining

syntypes catalogued as ANSP 355044) is about 22°, but this specimen is probably a juvenile, this value is close to the angle of the adult specimen of *M. martarum* sp. nov. (20°).

*Melanella martarum* sp. nov. can be easily distinguished from *Melanella cumingii* (A. Adams, 1854), known from Rapa Nui and other areas of the Indian and Pacific Oceans (*Osorio, 2018*). The straighter, thinner shell with almost flat whorls in the teleoconch of *M. martarum* is much different from the opaque and thicker shell with convex whorls of *M. cumingii*. The latter also reaches greater dimensions (around 20 mm long) (*Severns, 2011*).

*Melanella martarum* sp. nov. shares with *Melanella persimilis* (Kuroda & Habe, 1971), known from the upper slope of Japan (150–250 m) (*Hori & Matsuda, 2017*: pl. 100, fig. 2), the flat outline of teleoconch whorls. However, the former is relatively smaller (13 whorls, 8.1 mm *vs.* 15 whorls, 20 mm), having a narrower, more pointed apex, more convex base, and laterally expanded aperture.

*Melanella martarum* sp. nov. could be distinguished from *M. micans*, a shallow water species (up to 55 m) known from the Northeast Pacific (*Abbott, 1974*; *McLean & Gosliner, 1996*), mainly by the shape of the outer lip and expansion of the aperture. The former has a maximum projection of the outer lip in the middle height, while the latter has its maximum projection close to the distal area. The aperture of *M. martarum* sp. nov. is strongly expanded laterally in comparison to the almost straight profile of the outer lip of *M. micans* as seen in the frontal view. Besides that, the protoconch of *M. martarum* sp. nov. has more convex whorls than *M. micans*.

*Melanella martarum* sp. nov. differs from *Melanella martinii* (A. Adams, 1854), from Indo-West Pacific and is known from depths up to 30 m (*Hori & Matsuda, 2017*: pl. 98, fig. 9), and from *Melanella major* (G.B. Sowerby I, 1834), from Japan and Tropical West Pacific known from depths up to 10 m (*Hori & Matsuda, 2017*: pl. 98, fig. 10), by being smaller (maximum length 8.1 mm *vs.* ~35 mm in *M. martinii* and *M. major*), with straight apical whorls whereas it is curved in the others, and with sutures not well-impressed which is deeply demarcated in the others. Besides that, *M. martarum* sp. nov. can be distinguished by the sinuous outer lip compared to the more rectilinear outer lip of *M martini* and *M. major*.

*Melanella martarum* sp. nov. shares the almost flat teleoconch whorls with *Haliella chilensis Bartsch, 1917*, a deep-water species from off Chile, but can be easily distinguished by the acute apex in comparison to the dome-shaped protoconch of the latter species (*Bartsch, 1917*: pl. 43, fig. 6), a common feature of *Haliella* Monterosato, 1878.

*Melanella martarum* sp. nov. resembles *Polygireulima rutila* (Carpenter, 1864), known from the northeastern Pacific occurring in depths up to 658 m (*Bartsch, 1917*; *Abbott, 1974*). These species share the flat outline of teleoconch whorls, an elongated spire, and a similar spire angle. However, the former has a slightly faster increase in diameter, the base is more convex but could be distinguished mainly by the aperture more expanded laterally. The outer lip of *M. martarum* sp. nov. advances earlier, just below the suture, reaching its maximum projection in the middle height. In *P. rutila*, the outer lip starts in a straight profile and reaches its maximum projection below the middle of its height (see Vanatta 1899: figs. 5–6; images of holotype USNM 14828 of *P. rutila*: http://n2t.net/ark:/65665/m3a68bd2aa-cc7a-4973-bdd5-83e4e8674a56). The largest

specimen of *M. martarum* (SCBUCN-5482) with a similar number of whorls (∼13 whorls) to the holotype of *P. rutila* reaches a considerably larger dimensions (8.1 × 2.4 mm *vs.* 6.8 × 1.9 mm).

**Phylogenetic relationship of *Melanella martarum* sp. nov.**

We successfully sequenced COI, 16S rRNA, and H3 genes of a single specimen of *M. martarum* sp. nov. The final alignment of the COI resulted in 630 bp, H3 of 314 bp, and the 16S rDNA of 419 bp, and these concatenated sequences produced a final alignment of 1,363 bp. COI presented the greatest intragenus variability, with only an 81.9% similarity with the *M. acicula* sequence. H3 and 16S rDNA presented an identity percentage of 97.0% and 91.4% with *Melanella* sp. CKC-2011, respectively. The identity for H3 and 16S was slightly lower with the *M. acicula* sequences (96.3% and 91.9%), although within the expected percentage identity values between species of the same genus. Consistent identity values between markers are shown in the phylogenetic reconstructions (Fig. 5 and Fig. S1). Both the phylogenetic inferences calculated from the COI and with the 3 concatenated markers showed low bootstraps value in the deepest nodes, not being able to separate between different genera (Fig. 5 and Fig. S1). *Melanella martarum* sp. nov. clustered with other *Melanella* species identified in previous studies and was retrieved as the sister taxon to an unidentified *Melanella* (Fig. 5), which was not illustrated in the original publication (*Churchill, Strong & Foighil, 2011*), hampering further comparisons. In the phylogenetic reconstruction carried out with COI (Fig. S1), the sequences of the *Melanella* species do not present monophyly (it should be noted that the species reported as *Balcis eburnea*, the accepted name is *Melanella eburnea* Megerle von Mühlfeld 1824). However, the cluster formed between *M. martarum* and *M. acicula* is maintained in both phylogenetic reconstructions (Fig. 5 and Fig. S1).

Taking into account these results, the detailed phylogenetic relationship among congeners is premature, since the genus currently includes more than 200 species, and just a few sequences are available for a species-rich group such as Eulimidae.

## DISCUSSION

The type series of *Melanella martarum* sp. nov. is represented by specimens at different growth stages, varying from specimens with four to nine teleoconch whorls and lengths between 3.76–8.14 mm. The young individuals have an angulated ultimate whorl and an aperture with a rhomboid shape (Figs. 1F–1H), and the largest specimen (Figs. 1D–1E) shows a more rounded outline in the base and aperture. These ontogenetic differences are reasonably common in eulimids (*Lyons, 1978*: 81; *Bouchet & Warén, 1986*: 310; *Souza et al., 2018*: 926). The shape and dimensions of the protoconch are quite similar in all specimens studied, and some of them were collected in the same host. The seven eulimids were collected on four holothuroids, but there is no register of the specific individuals from each host.

Based exclusively on the shell morphology, a primary association of the newly described species with the genus *Melanella* could be inferred, despite the plasticity or diverse forms currently included in the genus. *Melanella martarum* sp. nov. has a conical, straight spire,

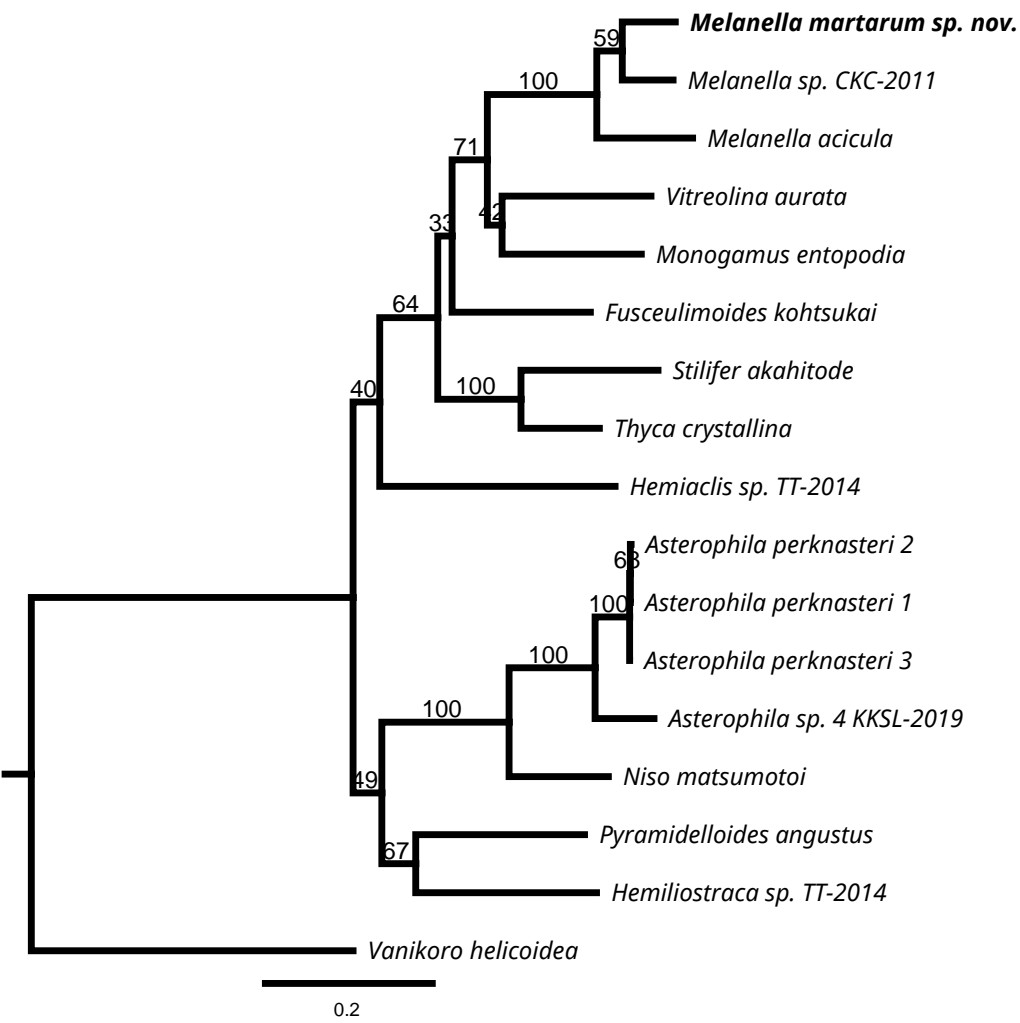

**Figure 5** **RaxML inferred phylogenetic reconstruction.** Based on 1,363-bp alignment of partial 16S, COI and H3 concatenate sequences of Eulimidae (bootstrap = 1,000). Bootstrap values obtained are shown in each node. *Vanikoro helicoidea* (Vanikoridae) was used as an outgroup.

with a sinuous outer lip, more similar to the species historically included in *Polygireulima* Sacco, 1892. The type species of *Polygireulima* is a fossil taxon and no data is available regarding the parasitic association, thus we follow *Souza & Pimenta (2019*: 429) in the broad concept of *Melanella* with a straight spire, differing from the strongly curved shell of *Melanella dufresnii* Bowdich, 1822 (type species of *Melanella*) (see *Souza & Pimenta, 2019* for details). Another possible generic placement would be *Eulima* Risso, 1826, which is also a "catchall" genus (*Bouchet & Warén, 1986*; *Hoffman & Freiwald, 2020*). *Eulima* comprises species mainly with flat teleoconch whorls and elongated shape, as *M. martarum* sp. nov. However, *Eulima* s.s., considering the type species *Eulima glabra* (da Costa, 1778), has a colored shell with brownish spiral bands, a narrower aperture, and a more straight outer lip (*Warén, 1989*). Hosts known for *Eulima* species are ophiuroids (*Bouchet & Warén, 1986*).

*Melanella acicula* (A. Gould, 1849), with a close relationship based on present data, has a similar shell morphology following the features cited previously. The lack of illustrations of *Melanella* sp. (CKC-2011) prevents us from checking the shell morphology of this taxon. Assessing with molecular data a broad number of *Melanella* species with these different patterns of straight and curved spires would be interesting for phylogenetic reconstruction and to check about this variation within the genus. Besides the shell morphology, the type of host also contributed to the generic classification, since *Melanella* comprise species parasitic on holothuroids (*Warén, 1983*; *Souza et al., 2018*). Despite being collected by an Agassiz trawl, *M. martarum* sp. nov. reached the surface still attached to its host, which is difficult with this type of sampling, especially in the deep sea. *Melanella martarum* sp. nov. possibly remains strongly attached to the host through the proboscis. Furthermore, the clustering of *M. martarum* sp. nov. with other *Melanella* in the molecular analysis, reinforced our previous assumption, despite the scarcity of molecular data on the genus. Thus, our generic classification could be supported by different types of data, although in the phylogenetic reconstruction with the COI *Melanella* was polyphyletic. The lack of monophyly of *Melanella* calls into question the usefulness of the morphological characters used to describe the genus, which has been discussed previously (*Bouchet & Warén, 1986*; *Souza & Pimenta, 2019*).

Mollusca endemism is usually high on seamounts (*Herrera et al., 2023*). In the seamounts of the Nazca and Salas & Gomez ridges, 96% of the gastropod species of the family Turridae and 25% of the Septibranchia bivalves are endemic (*Parin, Mironov & Nesis, 1997*). Since *M. martarum* sp. nov is the first record of the Eulimidae; little is known about the rate of endemism of species of this family in Salas & Gomez Ridge. But, on the bathyal slopes of the Azorean seamounts, 38.6% of Eulimidae species (17 of 44) were found to be endemic (*Hoffman & Freiwald, 2020*).

Most Eulimidae species found in Rapa Nui island are considered endemic. A few extend their distribution to other polynesic islands such as Hawaii, Cook, and Tuamotu (*e.g.*, *M. aciculata*) and only *M. cumingii* has a broad distribution from West Africa to Hawaii (*Osorio, 2018*; *Osorio, 2023*). The new species does not resemble them morphologically, and the available information indicates that it is only found on seamounts in SGR, which is home to a fauna characterized by high levels of endemism (*Friedlander et al., 2016*). However, this species probably occurs in surrounding areas due to the possible planktotrophic development of *M. martarum* sp. nov., inferred by the multispiral, conical protoconch. Host specificity and habitat preferences are not well known for a better comprehension of the distribution.

## CONCLUSIONS

*Melanella martarum* sp. nov. is the first gastropod of the family Eulimidae reported for seamounts of the Salas & Gómez ridge, an area with a high level of endemism but still poorly explored. The new species has only been reported from the summit of the seamount "Pearl" (Zhemchuznaya) (Lat. −25.59, Long. −89.13), but further sampling is needed in seamounts of the Salas y Gomez Ridge to have a clearer understanding of the geographic

distribution of *M. martarum* sp.nov. Despite the scarce molecular data for the family Eulimidae, the phylogenetic reconstruction allowed us to verify the assignment to the genus *Melanella* of this new species. However, the generation of more molecular data is necessary to clarify the taxonomy of the family Eulimidae at the species level.

### Abbreviations of other repositories

| | |
|---|---|
| **ANSP** | Academy of Natural Sciences of Philadelphia at Drexel University, Philadelphia, USA |
| **MCZ** | Museum of Comparative Zoology, Harvard University, Cambridge, Massachusetts, USA |
| **NHMUK** | Natural History Museum of the United Kingdom, London, United Kingdom. |

## ACKNOWLEDGEMENTS

For their assistance at sea, we would like to thank the Captain and crew of R/V Mirai, and the scientific personnel participating in the EPIC cruise. Special thanks to Erin Easton, Ariadna Mecho, Jan Tapia, and Jorge Avilés for their help during the collection, handling, and curation of the specimens. Special thanks also to Andrea Varela for helping with the lab work and Maria S. Romero for helping with the SEM. Thanks to the ANSP, MCZ, and NHMUK staff for providing images of the type of material, and to Omar Ojeda (UNAM, Mexico) for sharing images of other types under study.

### Funding

This study was funded by ANID- Millennium Science Initiative ESMOI and ANID-ATE 220044 BiodUCCT, FONDEQUIP EQM150109, and FONDECYT 1181153. The funders had no role in study design, data collection and analysis, decision to publish, or preparation of the manuscript.

### Grant Disclosures

The following grant information was disclosed by the authors:
ANID- Millennium Science Initiative ESMOI.
ANID-ATE: 220044 BiodUCCT.
FONDEQUIP: EQM150109.
FONDECYT: 1181153.

### Competing Interests

The authors declare there are no competing interests.

### Author Contributions

- Leonardo Santos de Souza performed the experiments, analyzed the data, prepared figures and/or tables, authored or reviewed drafts of the article, and approved the final draft.

- Cynthia M. Asorey conceived and designed the experiments, performed the experiments, analyzed the data, prepared figures and/or tables, authored or reviewed drafts of the article, and approved the final draft.
- Javier Sellanes conceived and designed the experiments, analyzed the data, authored or reviewed drafts of the article, and approved the final draft.

### Ethics

The following information was supplied relating to ethical approvals (i.e., approving body and any reference numbers):

Under-secretariat of Fisheries and Aquaculture of Chile (SUBPESCA).

### Data Availability

The melanella martarum sequences are available in the Supplementary File and at GenBank: OP589975, OP575953 and OP577852.

### New Species Registration

The following information was supplied regarding the registration of a newly described species:

Publication LSID: urn:lsid:zoobank.org:pub:716D01AC-7DA7-4B2D-A4D5-0DD925825BC6

Melanella martarum sp. nov. LSID: urn:lsid:zoobank.org:act:CC4C4BE5-4D7F-4EF3-8926-9B960C01C3CA

### Supplemental Information

Supplemental information for this article can be found online at http://dx.doi.org/10.7717/peerj.16932#supplemental-information.

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
