# Peer review of "Melanella martarum sp. nov. (Gastropoda: Eulimidae): the first parasitic deep-sea snail reported for the Salas & Gomez Ridge"

_PeerJ, doi:10.7717/peerj.16932_

## Round 0.1 · original submission · Major Revisions

The manuscript received comments from two reviewers and one review is awaiting. However, I suggest Major revision as per two reviewers' comments. The second reviewer has pointed out to compare your species with its congeners extensively. Please follow their comments carefully and revise the manuscript accordingly with your rebuttal.

·

Basic reporting

The text is very well written and clear. I have suggested very minor modifications. The literature seems complete to my knowledge, but some references are incomplete. I have highlighted those. Genus and species names need to be in italics throughout the manuscript, use of comma in references should be checked as there are some inconsistencies.

Experimental design

no comment

Validity of the findings

The manuscript reports a new species of the eulimellid gastropod genus Melanella. ICZN requirements are met. I am, however, not completely convinced that only one species is present here. Paratype SCBUCN 5482 (Figs 1D, E, I-K), used also for genetics, looks quite different from the rest, with a much more rounded aperture and base. Table 1 shows it to have almost double the length of the other specimens, but this seems not clear from the plate. I did not measure the scales though. Maybe one or two sentences in the discussion can make this easier to understand. The diagnosis/description could benefit from adding other characters like spire angle, because species in this family are very similar. A few are used to separate the new species from others.

·

Basic reporting

English is clear, concise.
References need to be updated to include similar seamount areas.
Thorough checking of references is required; both formatting mistakes and omissions have been found
Data sharing is appropriate

Experimental design

Research fits in PeerJ; a limiting factor is that we deal with a single species only.
Research is well defined and meaningful even though as a whole a small incremental step is achieved in the knowledge of Eulimidae.
The investigation is rigorous and professional.
The analytical results needs to be extended to look wider into the genus to ensure that we deal with a new Pacific species.
Methods used can be replicated easily.

Validity of the findings

The raw data is of good quality.
The morphological analysis of results needs to be extended to include a wide set of Pacific species to ensure that a new species is confirmed.
The molecular analysis needs extension to a larger set of species using COI only. The current concatenated sequence limits the phylogenetic tree to few species only. The added value of 16S and H3 is limited.
Having done this then the general conclusions can be adequately supported

Additional comments

Generally an acceptable manuscript with the description of a single species from a combination of morphology and molecular sequences. The type material, the type descriptions, type imaging are adequate, the genetic analyses is extensive with three genetic sequences in multiple specimens.
Currently, the comparative material is oddly chosen and the comparison with congeneric species needs to be drastically extended.
The genetic analysis requires broadening by adding a separate analysis on more congeners using COI only (available in Genbank).
The discussion needs mentioning of recent and historic molluscan endimicity on seamounts in general and those regarding eulimids in particular.
The references in the text and the formatting and order in the bibliography needs careful checking. I discovered two references not incorporated in the back; there are possibly more.
I wish you success in correcting and re-submitting your manuscript.
With kind regards, Leon Hoffman

Detailed findings
Abstract
Line 44 remove ‘usually’, outside the depth range stated there is no habitat for Eulimidae
48 Agassiz trawl, please complete the gear
49 25.59°S and 89.13°W rather than the strange coordinate location? You realize that the accuracy is about 1 km2 which is inaccurate.
59 and 61 ‘y’ is ‘&’ elsewhere, both are correct, please pick one.
Keywords
68 why ridges and not Ridge?
Introduction
83 suggest to use consistently Warén with accent
98 generalist? Singular?
99 eulimid? With lower case, only family and genus with capital start.
107-108 These expeditions (plural) as you stated it earlier?
109 Within the CEEZ? In 106 you stated that they were outside CEEZ. It also comes back in 114, quite confusing. Can we get rid of the CEEZ altogether or is this a hot national item?
109 the reference CONA, 2016 is missing in the back
M&M
129 location notation as above.
140-143 consider removing the primer sequences and you have them named and referred to.
Description
198 height (and width) protoconch?
202, 213 please use ‘ultimate’ rather than ‘body’ whorl. The animal extends well beyond the last whorl inside.
214 outer lip not thickened; are these all subadults?
218 Why choose a subadult specimen as holotype when you have a live-collected large specimen (the last paratype) with a perfect lip: a specimen that probably comes closest to the adult features?
219-220, 242-243 coordinates in common format stated earlier.
230 It is unclear why Melanella aciculata was chosen as comparative material. Unlike the new species, M. aciculata (and acicula) is from shallow water and there are many congeneric species that are similar to the new taxon. If you have done so for molecular reasons (there are COI results for this species in Genbank) then I would also expect M. thersites and M. lubrica (all in Genbank with COI).
The same question for M. acicula. There is a full set of genomes in Genbank (16S, 28S, H3 and COI) but this species was not imaged in the manuscript (good image in WoRMS!)?
240 any specific location on the host: how attached (proboscis or other)
253 the statement “a consequence of….” is speculative to me
256 as you make a distinction in the spire angle please measure it, include it in the description and give a value for acicula and aciculata and show the difference
257 I am afraid you never found adult specimens and the absence of callus inside the inner and outer lip could well be an age issue.
261 Can you measure the apical angle of acicula?
271 Why compare with a Haliella when there are many more Melanellas?
How about Melanella medipacifica (very similar, with aligned scars and thickened lip), martini (fat, curved, convex whorls, aligned scars), M. grandis (twisted apex, aligned scars), M. major, M. perplexa, M. hemphelli, etecera. Did you make an inventory of congenerics in the Pacific (from the WoRMS list). Then you could make a sub set of only upper bathyal species and then you make a comparison of all remaining species. Alternatively, select all straight species with flattened whorls from the Indo-pacific and compare.
In short, I am unhappy about your choice of species for comparison; look for a similarity in habitat (including depth) and morphology. This way I cannot judge whether your species has been described or not.
Phylogeny
276 You base your molecular differentiation on a concatenation of 3 genes. You show that the differentiation is best in COI, and in Mollusca, this is mostly the case. By only investigating the concatenation, you get only M. acicula in your tree. By taking only COI (superior diffrentiation anyway) you can also include M. aciculata, M. thersites and M. lubrica and many many more eulimids in your tree (all available in Genbank). First you would be more convincing in choosing Melanella. Second, you may say something about the clustering of the genera (or the lack thereof).
You provide differences between species; I would like to get a quote for the variation within the new species for COI and for the concatenated sequence. Then show that the in infraspecific variation is much smaller than the interspecific variations; do this at least for COI.
Discussion
296 The first sentence is a bold statement, a fit within Eulima (also polyphyletic) would also be feasible (refer to Bouchet & Warén, 1986; Hoffman & Freiwald, 2020).
324 Friedlander et al, 2016 is not in the reference list. I have not heard of Friedlander but there are better references of endemic rates of molluscs in general (Puillandre, Marshall, Caballero Herrero et at 2023) and eulimids on seamounts in particular (Dautzenberg & Fischer, 1896, 1897, Bouchet and Warén, 1986, Hoffman & Freiwald, 2020)
First, recheck carefully that all references in the text are enclosed in your bibliography and that that all items in the bibliography are referred to.
Second, I wish a better discussion on endemics on seamounts and on eulimids in particular referring to important recent and historic papers.
333 coordinates as above. Format of Salas y Gomez
References
399, 402 Recheck your formats of the references carefully. Here are two from the same author with different formatting; I believe the last is correct.
433 Recheck the order, I believe this Takano & Kano needs to go after Takano & Goto above?
Figures
Caption Figure 1 errors: surfac, suturee
Figure 2 In my view G-H is a different species. A-F have a straight spire with central columella and aligned outer lip. G-H is fatter, twisted spire, an eccentric columella with a non aligned lip.

---

## Round 0.2 · accepted · Accept

Two reviewers have accepted all your revisions for their comments and suggestions. Hence, based on those reviews, I recommend the acceptance of this article. However, still few minor corrections recommended by the reviewers need to be carried out.

·

Basic reporting

I have reviewed this manuscript before and find that this new version responds well to my previous comments. Usage of more characters and comparison with more taxa makes the discussion much better than before. I annotated very few remaining details on the pdf file for consideration.

Experimental design

This is a corrected manuscript. I have no additional comments

Validity of the findings

This is a corrected manuscript. I have no additional comments

·

Basic reporting

This is the response following a review and re-submittal by the authors. For earlier comments I refer to the first review.
The text is clear, concise and logical, fit for publication.
The literature was updated, corrected and more complete in the current version. Make sure that the full set is checked for the Peer-J formats; use italic font for all generic and specific taxon names. Occasionally, in the corrections a space is missing.
Quality of tables and figures is fit for publication.

Experimental design

An analysis of COI only was added, I thank the authors for doing this; the supplementary file on this is adequate.

Validity of the findings

no comments in addition to previous version and corrections provided.

Additional comments

I thank the authors for their attention to all comments provided and wish them success in the wrapping up